# Sun-Safe Zones: Investigating Integrated Shading Strategies for Children’s Play Areas in Urban Parks

**DOI:** 10.3390/ijerph20010114

**Published:** 2022-12-22

**Authors:** Nikhil C. Cherian, Chamila Subasinghe

**Affiliations:** School of Design and the Built Environment, Curtin University, Perth, WA 6102, Australia

**Keywords:** children’s play areas, urban parks, shade, canopy cover, UVR exposure

## Abstract

Although Open Space Ratio is a critical control in the Development Approval process, there are no ultraviolet radiation (UVR) protection guidelines for urban parks. This study explores key strategies for shade provision in children’s play areas in urban parks, aiming to promote sun-safe play environments against alarming skin cancer trends. The literature review identified primary issues affecting UVR exposure in public venues, and the research comprises a shade audit of Beaton Park in Dalkeith. The methods involved using virtual park modeling and Shadow Analysis simulations to generate the daily average number of hours in shade for each month. Our recommendations based on this analysis are (a) a minimum canopy cover representing 50% of the entire ground cover; (b) a minimum diameter for a shade (umbrella) of about 2.5 times the diameter of the table; and (c) an ideal umbrella height of 90 cm from the table surface. This research proposes a potential nexus between landscape design and a UVR protection framework for child-friendly Sun-safe Zones (SsZ).

## 1. Introduction

Australia has one of the highest skin cancer rates in the world [1,2]. Australia’s geographical proximity to the equator subjects the country to high levels of UVR [3]. Being the sunniest city in Australia, with about 3200 h of annual sunshine (Figure 1), Perth is continually bombarded by hazardous levels of UVR [4]. This UVR exposure remains the principal cause of skin cancer in the country, especially in Western Australia (WA) [5]. Providing adequate shade in public spaces has lasting implications in addressing this issue.

Skin exposed to high levels of UVR poses a significant risk of damage [6]. Research suggests that the skin is most susceptible to sunburn and immediate damage in toddlers. This predisposition is attributed to the underdevelopment of melanin, the natural skin pigment responsible for sun protection during infancy [7]. Excessive sun exposure in early childhood can significantly elevate the risk of contracting skin cancer in the future, making children the most vulnerable group to UVR-induced skin damage [7,8,9]. Additionally, research indicates that children are more susceptible to heat stress owing to their higher body surface area to mass ratio, underdeveloped body regulatory mechanisms, higher metabolic activities, and faster heart rates compared to adults [10,11]. These factors make children more vulnerable to heat stress induced by extreme temperatures.

According to Cancer Council WA [12], Perth is subject to only fifteen days annually on which the Ultraviolet Index (UVI) is under three, i.e., the UVR level above which sun protection is recommended. Such exposure underlines the urgency of public venues offering adequate shading throughout the year and further establishes the necessity of prioritizing shade in spaces frequented by children. Furthermore, shade is necessary for children’s play areas, as the exposed play equipment tends to become dangerously hot in the summer, which can pose grave hazards to children [13].

Holman et al. [14] indicate that good-quality shade significantly mitigates overall UVR exposure, defining shade as “a built or natural intervention that provides protection from ultraviolet radiation” (p. 1607). According to Parisi and Turnbull [15], efficiently designed and quality shade must account for: (a) reflected UVR from the surrounding environment associated with the albedo (i.e., reflectivity) of the ground surface materials, in addition to the climatic context under consideration (Figure 2), (b) the shade structure’s dimensions, shape, orientation, and sky exposure, and (c) characteristics of the shade provided, such as whether it is by natural means or via the built structure, the properties of the shading material for built structures or canopy density of vegetation [14].

Therefore, resultant UVR exposure is a consequence of the type and implementation of shade (its characteristics) and the level of UVR protection dictated by the amount of cloud cover and the physical features of the site in question (the reflectivity of surfaces) [16].

There have not been many published studies exploring UVR exposure and shade availability in children’s play areas. The two most comparable studies are: (a) the “PlaSMa” study by Schneider et al. [17], which investigated 144 playgrounds in Mannheim, Germany, in order to establish a significant lack of shade in play areas via quantitative research methods, and (b) a study by Gage et al. [18] across 50 New Zealand playgrounds that investigated the quantitative and qualitative shade availability in Wellington play areas, revealing insufficient shade. However, to the best of our knowledge, no prior studies have addressed both quantitative and qualitative shade availability in a children’s playground setting in Australia, or proposed recommendations around enhanced UVR protection.

As such, the primary contributions of this paper are the following:(a)We conduct a park-usage analysis and shade audit of Beaton Park, Dalkeith, to analyze its shade availability both quantitatively and qualitatively.(b)We identify the issues in shade quality and propose recommendations for integrated shading strategies, which can offer direction for effective shading of children’s play areas to maximize UVR protection.

### Literature Review

The albedo value of a surface is an essential factor to consider when designing effective shading, as the reflected UVR may cause considerable skin damage even under shaded settings [13]. The greater the albedo value of a surface, the greater the amount of solar radiation reflected off the surface and the less absorbed. This means that, all else being equal (such as thermal admittances), a surface with greater albedo warms little compared to one with lower albedo, as the latter absorbs most of the light incident upon it. A surface with a lighter appearance has a high albedo and reflects most of the incident light without changing temperature. In contrast, a surface with a dark appearance has a lower albedo and absorbs most of the incident radiation (Figure 3). Pigment colorant researchers are developing infrared-reflective pigments that induce a darker surface to reflect near the infrared wavelength of the electromagnetic spectrum (Figure 4) similar to a light surface [19]. While this technology should result in much cooler surface temperatures, its impact on reflected UVR remains to be established.

Research indicates the surface temperature difference between shaded and unshaded asphalt to be as much as 22 °C [20,21]. Commonly, surfaces with hard or smooth textures indicate a much higher albedo (and thus reflect more UVR) compared to natural ground cover with softer and varied edges, such as grass [15,22]. For this reason, Parisi and Turnbull [15] recommend using natural ground cover as much as possible (or other surface materials with low albedos) under shade structures as a substitute for higher-albedo surfaces, such as concrete pavers. Pfautsch et al. [23] examined the importance of shade in preventing playground burns due to overheating of floor surfaces. They found that synthetic turf was one of the hottest surface materials, whereas natural turf was one of the coolest. The authors established that shaded floor surfaces exhibited no meaningful variances in surface temperatures across color tones and materials [23].

According to Madden et al. [11], thoughtfully designed children’s play areas facilitate microscale cooling, thereby delivering much-needed refuge in extreme summer months. Poorly designed play spaces with unsuitable ground surface materials induce micro-heat island effects by raising the air temperature, creating unsafe urban environments for children’s play. Such hazardous play environments discourage parents from providing outdoor play opportunities for their children; in turn, parents promote sedentary activities related to the increased time spent indoors [11].

Skin-damaging UVR remains subliminal to human sensory perception, unlike the heat perceived from infrared radiation (Figure 4) [24]. This inability to perceive UVR can create a false sense of security in users under shade, misleading them into believing themselves to be safe from UVR exposure and consequently prolonging their stay or dropping their guard. These considerations highlight that effective shading may not be as simple or straightforward as initially supposed.

The dimensions of shade structures are another factor that considerably impacts UVR protection. Parisi and Turnbull [15] conducted research with shade structures of diverse sizes and found that the protection factor (PF) offered by the structures decreased as the solar zenith angle (SZA) increased (which results in a lower elevation angle, as seen in Figure 5). Thus, their findings established a direct correlation between the amount of indirect UVR and the sun’s angle. Furthermore, their research highlighted the importance of accounting for lower sun (elevation) angles when designing structural (built) shade to effectively mitigate direct and indirect UVR. A study by Parsons et al. [25] demonstrates the same. Parisi and Turnbull [15] established that another significant parameter that dictates the quality of the built shade is the amount of diffused UVR permeating the shade structures from its exposed sides.

The above visualization scene (Figure 6) of the Baldivis One71 Pump Track, WA, demonstrates how the shade cast by the structure does not fall directly beneath it, consequently exposing the seating to solar radiation. The scene is set for the southern winter solstice (21 June) at 12:00 pm (a time of peak UVR); it illustrates how the lower elevation angle of the winter sun dictates the position of the shade cast, and consequently also the protection factor of the structure. Therefore, it becomes imperative to have roof overhangs or integration of solar protection on the sides of the shade structure to mitigate the amount of diffused and scattered UVR pervading it [26]. This lateral protection can be efficiently achieved by means of trees and vegetation (Figure 7).

Landscape shading via vegetation is a vital element in providing efficient shade in urban parks and children’s play areas. According to Zarr and Conway [27], the environmental and public health benefits that nature-based shade interventions confer establish the need to prioritize shade trees over built shade structures.

The PF offered by shade trees is dependent on several factors, including canopy density, height of the canopy, season, SZA of the sun, and the amount of cloud cover [15]. The PF naturally varies for deciduous trees or those that indicate variations in their canopy densities with the change of seasons. However, research has shown that denser tree canopy offers higher UVR protection; thus, the overall PF can be maximized by planting a cluster of trees with thick canopy cover (Figure 8) or locating them close to built shade structures [26]. Parsons et al. [25] demonstrated that “trees with dense foliage and low branches, such as mango and Chinese elm, gave the most effective UV protection ratio” (p. 329) as compared with trees more commonly found in Australian recreational spaces, such as eucalyptus or Norfolk Island pine. Effective UVR protection (i.e., PF > 15) was observed in densely forested areas and below widely overhanging structures [25]. The research by Parsons et al. [25] additionally suggested that the variations in the shade offered were not significant enough to endorse dependence on any specific tree species. Research by Brown et al. [28] indicated that the shade afforded by trees is the most efficient cooling strategy concerning urban parks and children’s play areas, both because of its ability to shield against penetrating UVR and also owing to its evaporative cooling properties. However, in order to maximize its shade benefits, Brown et al. [28] suggested that it is crucial to consider the shading performance of foliage types when designing urban parks. Their research indicated that canopy density has a considerable impact on UVR protection as well as on the thermal comfort offered during summer conditions. It is crucial to design landscape shading in the ways that are most appropriate to the climatic context under consideration while factoring in seasonal variability. For instance, it is logical to maximize shade in climatic contexts with scorching summer months and minimal seasonal variance. However, in climatic zones resembling that of Perth, where the winter months are characterized by partially cloudy days and would benefit from some exposure to UVR, it is best to incorporate a balanced proportion of open landscaped patches and zones abounding with leafy shade trees with generous canopies [28].

The comfort of users is another critical factor that governs the utility of shade in public spaces. However, the ambient temperature that provides thermal comfort is not a reliable indicator of UVR exposure and its intensity, due to the prevalence of diffused UVR. This inability to perceive UVR makes it possible to experience skin damage from overexposure to sunlight even in colder conditions [14]. This behavior of users becomes problematic for public health in the summer months, during which users tend to wear less clothing, exposing themselves to more UVR [15]. The provision of comfortable seating under a shaded setting promotes the use of that shaded space, especially during the summer when it is most likely to be used [14,29].

Consequently, adequate shading encourages the use of outdoor spaces, whereas the lack of it leads to the disuse of such spaces altogether. Moreover, park users tend to prefer areas that provide warmth during the winter months [15,30]. Holman et al. [14] suggest that the design of such “warm shade” may be achieved via the “use of polycarbonate and laminated glass with UV protective coatings” (p. 1609). Moreover, implementing novel shade interventions such as kinetic or ephemeral shade structures could enhance shade efficacy by accounting for seasonal variations. Such interventions should effectively mitigate overexposure during the summer months while allowing some sun exposure during the winter, in this way permitting essential UV-induced Vitamin D synthesis [13]. Nevertheless, it is crucial to understand that shade does not offer 100% protection from solar radiation; for this reason, the use of personal protective measures, such as sunscreen and sun-protective clothing, remains essential when using public venues [31].

## 2. Materials and Methods

An online survey called *Shade Stories* was organized by Cancer Council WA [12] to generate data on the community perception of shade availability at public venues across WA in order to identify specific venues where the community deemed shade as being “good, bad and everything in between” (para. 1). We selected Jo Wheatley All Abilities Play Space (Beaton Park), Dalkeith, WA, Australia, as the primary case study, based on the survey results. The venue was selected on account of its being among the better-shaded urban parks in WA according to public perception. Three visual shade audits of the venue were conducted, between July 2020 and October 2020, to evaluate shade availability and identify the spaces frequented by patrons; data were recorded by pen and paper. This was followed by simulation studies to analyze the shade availability during various times of the day and times of the year. The simulation method involved virtual park modeling akin to that employed by van Vliet et al. [32] in their environmental preferences research. An analysis of the existing vegetation identified all the tree species present at the venue and investigated their expected heights and canopy spread at maturity.

The primary method of shade analysis involved generating digital 3D models of Beaton Park (using Google SketchUp Pro 2018 and Enscape 2.7.1) to conduct comprehensive solar studies via SketchUp’s Shadows feature, allowing a geolocated model to cast shadows according to changing sun paths. We utilized this feature to investigate the available shade during various times of the day and year. The solar analyses comprised observations made during the morning, solar noon (the time of peak UVR when the sun is immediately overhead), and late afternoon, i.e., within peak UVR exposure hours as identified by Stoneham et al. [31], which fall between 9.00 am and 3.00 pm in Australia. This study considers the winter and summer solstices (21 June and 21 December, respectively). As part of the analysis, our research employed an environmental simulation software called Shadow Analysis (2020), available as a SketchUp plug-in; for each space and each month of the year, the software generated the daily average number of hours in the shade. Based on the findings of the case study analysis, this study has proposed viable solutions in response to the issues identified.

### Case Study: Beaton Park, Dalkeith

This section comprises a park-usage analysis of Beaton Park, looking at key urban metrics to understand the scale and function of the park and the catchment area that it services (Table 1, Figure 9). We investigated the temperature history of the site (Table 2) and the materiality of ground surfaces (Figure 10) to understand their reflectivity (Table 3), and also examined the species of vegetation prevalent at the site (Table 4). Shade analysis was then conducted for the venue, which forms the crux of this investigation.

*Jo Wheatley All Abilities Play Space* (*Beaton Park*) in Dalkeith, covering over 20,000 m^2^, is planned around existing trees, with minimal disruption of the natural environment, and is designed for inclusivity. The playground comprises a sensory walk, inclusive picnic and BBQ zones, decks, ramps and slides, an inclusive ball games area, exercise equipment, a community garden with BBQ facilities, and designated sand and water play zones. The availability of amenities and the integration of inclusive infrastructures, such as accessible toilets and wheelchair-accessible spaces, make this facility the first of its kind in terms of scale across WA [33].

Table 5 comprises information related to the three shade audits of the venue that we conducted. Figure 11 provides essential information on how the spaces of the park were used by patrons, revealing the patterns of usage. Table 6 comprises a shade audit of the park, analyzing the canopy cover and the percentage of available shade. We used AutoCAD to generate the data by calculating the relevant footprints, determined under the assumption of the sun being directly overhead at solar noon and forming a corresponding shade area (such as under tree canopies). This information is supplemented by the data generated using Shadow Analysis (for SketchUp) to examine the total shading hours during both solstices. The analysis further reveals which areas of the park are in shade and for what durations. The implications of these data are discussed further in Section 3.

**Figure 11 ijerph-20-00114-f011:**
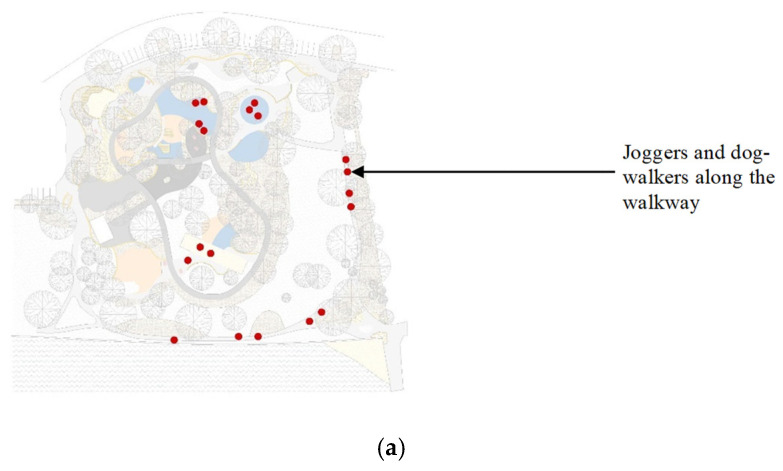
(**a**) Visit 1, (**b**) Visit 2, (**c**) Visit 3.

**Figure 12 ijerph-20-00114-f012:**
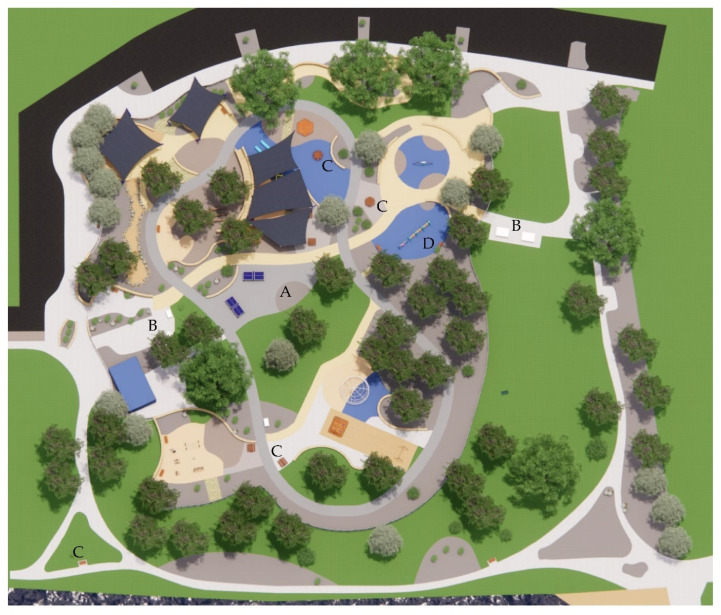
Canopy cover and shade at 12 pm on the summer solstice.

**Figure 13 ijerph-20-00114-f013:**
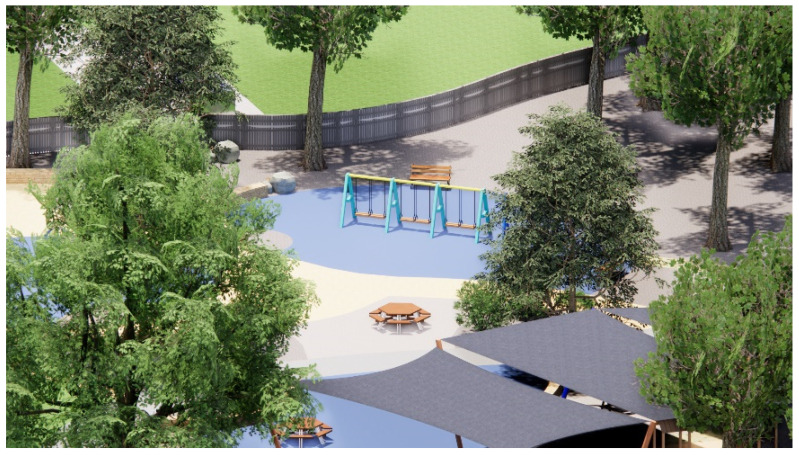
Shade sail, swings, and unshaded seating areas during midday on the summer solstice.


*Shadow Analysis simulations depicting the average number of hours (per day) in shade.*


**Figure 14 ijerph-20-00114-f014:**
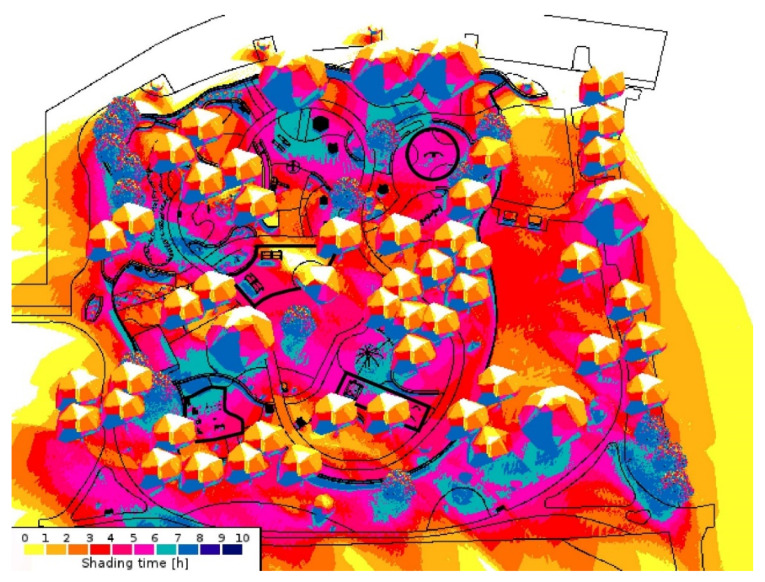
Winter solstice: 21 June (ten total daylight hours per day).

**Figure 15 ijerph-20-00114-f015:**
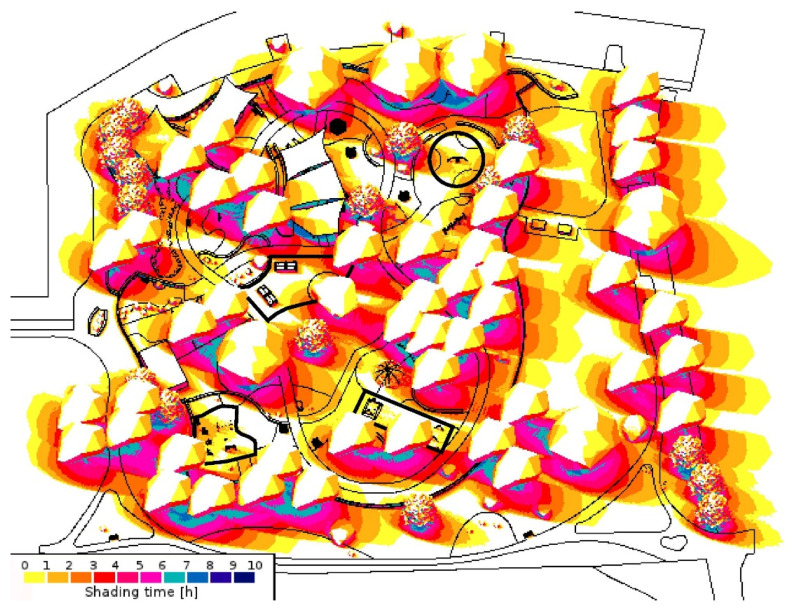
Summer solstice: 21 December (fourteen total daylight hours per day).

## 3. Results

Jo Wheatly All Abilities Playspace is one venue that the *Shade Stories* online survey results indicated to be among Perth’s better-shaded urban parks [12]. However, the shade audit reveals that public perception does not concur with objective measurement in this case. The quantitative analysis reveals the canopy cover to be only 27% of the entire ground cover (Table 6), which falls short of the recommendation of 30% by Stoneham et al. [31] of the Australian Institute of Environmental Health. Furthermore, the percentage of total shade provided during midday of the summer solstice is no more than 29% of the entire ground cover (Table 6), inclusive of the shade offered by the temporary shade sails erected over the play equipment and sand-play zones.

The aerial view in Figure 12 indicates that most spaces, such as the ball games zone with the ping pong tables (A), BBQ facilities (B), seating (C), and swings (D) are left completely sun-exposed at midday during the summer solstice, while shade sails are limited to a few play zones. The zone dedicated to sand-play is partially shaded by the shade structure, whereas the remaining area remains unshaded. The reflected UVR radiated by the sand becomes problematic in this case. Figure 13 depicts how the swings and the surrounding seating spaces are in full sun exposure during the midday at this time of the year. Results from the Shadow Analysis program reveal that most play spaces receive under 2 h of shade during the summer solstice, whereas they remain relatively well-shaded during the winter solstice, with around 4–5 h of shade (Figure 14 and Figure 15).

A sizable percentage of the park is covered with soft-fall rubber surfaces (the blue surfaces seen in Figure 16 are some of them). While these surfaces provide considerable protection from UVR owing to their low albedo value, they tend to become heated because of their high absorption [13,38]. Research by Vanos et al. [21] indicates that such surfaces can reach alarming temperatures of up to 82 °C in the summer. Such extreme temperatures pose significant health hazards to children’s safety, as research suggests that temperatures over 80 °C cause severe burns within seconds [39]. Between the hours of peak UVR from 9:00 am to 3:00 pm, the play zones accommodating the swings, ball games, and climbing structure, as well as the seating zone (intended for carers) between the flying fox and the exercise zone, are in partial shade (20–50%) only from 9:00 am to 9:30 am (Figure 17).

Similarly, Figure 18 indicates how the Flying Fox area (F), the climbing structure (C), and the intergenerational exercise equipment (X) remain sun-exposed at midday on the summer solstice. Apart from being in full sun exposure during the summer solstice, the Flying Fox poses the additional problem of being located on sand and consequently radiates a significant degree of heat and glare. In contrast, the climbing structure and the exercise equipment are located on soft-fall rubber surfaces, which can pose severe burn risks upon skin contact. Moreover, all the seating space surrounding the equipment, which is intended for seating supervising parents, remains unshaded.

### Recommendations

This section further discusses the case study findings and proposes recommendations in response to the issues raised. First, parents and carers frequently use seating spaces in children’s play areas as they supervise children engaged in play (Figure 11). Because of the fundamental purpose it serves, such seating spaces are likely to be used regardless of whether or not they are well-shaded. Therefore, in addition to shading all play equipment, there is a pressing need to appropriately shade all seating spaces surrounding the equipment. The ideal solution would be that all play equipment, BBQ zones, picnic tables, and seating should be under the shade of existing trees as much as possible.

Second, picnic tables integrating umbrellas can be an effective shading solution where natural canopy cover is lacking. The dimensions and positioning of the umbrella should be such that it provides at least 50% shade for the seating underneath during the winter solstice when the solar elevation angle is lower, and shadows fall obliquely. The user should be seated on the side of the table where the shadow falls to effectively utilize the shade (Figure 19). The shade provided for such a seating space is dependent on the diameter of the umbrella and its height from the table/ground surface. The larger the umbrella’s diameter and the lower the height of the shade, the larger the shaded area. Research through 3D modeling indicated that the diameter of the umbrella should be at least 2.5 times the diameter (or diagonal length, if hexagonal in shape) of the table. Furthermore, the umbrella should be positioned as low as possible; 90 cm from the table surface represents an ideal height (Figure 20).

The nature of ground surface cover is another vital factor that dictates UVR protection due to its reflectivity (albedo). While low albedo surfaces reflect the least amount of radiation, these can become dangerously hot because of their high absorption. Therefore, a solution is to prefer low-albedo surfaces (as these minimize reflected UVR) while avoiding dark-colored surfaces (such as black). However, these need to be sufficiently shaded during the peak hours of UVR to prevent overheating and ensure protection against the hazard of thermal burns. Sand, being a highly reflective ground surface, can be replaced with soft-fall mulch in locations not designated for sand play. Furthermore, it is preferable to replace soft-fall rubber with wood mulch, wherever possible, to mitigate the danger posed by the overheating of rubber surfaces.

Shade systems can lend character to large open spaces that otherwise appear expansive and overwhelming. Additionally, they can be useful in distinguishing discrete activity zones by creating implicit borders, thereby reinforcing the identities of disparate spaces [40]. However, it is essential to retain the outdoor atmosphere of openness desired in parks and children’s play areas as much as possible. For instance, it negatively impacts user experience to erect shade sails over swings or the Flying Fox equipment at Beaton Park. The shade concern of the Flying Fox can be solved by integrating shade into the equipment itself. A polycarbonate sheeting tinted with UVR protection, affixed to the poles connecting the zip-line, can provide the necessary shade during the sweltering summer months while maintaining the essential characteristic of openness that an opaque shading device fails to provide (Figure 21). However, it is crucial to provide at least a one-meter overhang on all sides of the equipment in order to maximize UVR protection (Figure 21). Furthermore, as already discussed, swapping the ground surface material from sand to wood mulch would reduce the exposure to reflected UVR.

The sensitive nature of children’s skin renders them highly susceptible to thermal burns in playground environments. This hazard becomes more concerning in the heat of the summer months when equipment tends to become overheated and unusable. Moreover, the summer heat presents the additional hazard of heat exhaustion among children [21]. Therefore, the design of the play environment needs to consider the perils of heat exposure to its users. The incorporation of shade trees becomes crucial when combating this problem, thanks to their ability to lower the ambient temperature and provide direct shade. Moreover, all play equipment, whether metal or plastic, should be treated with a heat-mitigating coating to reduce the risk of playground thermal burns.

The concept of children’s play should be reimagined when designing equipment to incorporate design strategies that integrate shade within play activity. This rethinking of play displaces the prevalent practice of simply erecting detached shade structures over play zones. Designing play spaces integrating mounds, tunnels, tubes, domes, and aerial walkways is a valuable strategy for ensuring that play areas offer adequate shade. For example, children’s play areas with expansive grassed areas can be integrated with mounds/tunnels covered with natural turf. Alternatively, the mounds can be constructed entirely from adobe or rammed earth (Figure 22 and Figure 23). The natural cooling properties of adobe can help to maintain an ambient temperature even in the hottest weather [41], whereas the material’s low albedo protects against indirect UVR. Such play objects offer opportunities for movement games among children, which facilitate social and cognitive play [42].

## 4. Discussion

The visualization of Tomato Lake Playground, Kewdale, WA, in Figure 24, illustrates an urban park with a much higher canopy cover. Estimating the vegetation footprint in AutoCAD suggests a canopy cover of 52% of landscape shade; nonetheless, it can be seen from Figure 24 that, during midday on the summer solstice, for which the scene in question is set, much of the play spaces remain in complete sun exposure. It can thus be deduced that even a canopy cover of 52% can become inadequate if the shade does not encompass activity areas, contrary to the recommendation of 30% by Stoneham et al. [31]. We, therefore, posit that a canopy cover of 50% of the entire ground cover is more appropriate, and even necessary as a minimum requirement, with the additional caveat that the cover be strategically placed around activity zones (as identified in Figure 11).

Research by Égerházi et al. [43] comprised a comparable microclimate study of a children’s playground in Szeged, Hungary. Apart from the obvious difference in climatic contexts, the investigation involved simulations of thermal comfort conditions using urban microclimate modeling; thermal considerations are one key aspect that is discounted from the present study. Research was conducted by Gage et al. [18] in a similar microclimate study to the current study. Their research was conducted across 50 randomly selected playgrounds in Wellington, New Zealand, to analyze shade availability, demonstrating a similar objective of UVR protection (suggesting a climatic context of similarly alarming UVR levels); the large sample size lends their study a high degree of scientific rigor. The methods employed for determining shade quality relied on the use of a UV solar meter, in contrast to the digital modeling involved in the current study. Further, their study accounted for variances in tree foliage density, unlike the current study [18]. Another comparable investigation, called the “PlaSMa” study, was conducted in the German city of Mannheim across 150 playgrounds, in what is “the biggest investigation of UV exposure on playgrounds worldwide” [17] (para. 29). The methods used in the study to calculate shade areas are remarkably similar to how we have used AutoCAD to estimate canopy cover percentage. However, the research methodology applied was restricted to quantitative research and did not account for shade quality.

There are several limitations to our study. Only three site visits were recorded, and these do not constitute a statistically significant data set to reliably establish any patterns of occupation associated with the venue. The shade audit does not consider the Leaf Area Indices (LAI) of the various tree species or their actual heights, instead being modeled based on the average tree heights and canopy spreads of the species. The methods used here would have been further improved if the digital modeling and simulations were supplemented by the methods employed by Gage et al. [18], which uses UV solar meters to estimate shade quality at the site. While albedos of the common ground surface materials were identified, the tangible effects of the albedo values on shade quality were not determined, as the model cannot account for multiple reflections. Moreover, the albedos of play equipment, due to color and materiality, were discounted. In addition, this study does not account for the impact of thermal comfort conditions on park usage, which was deemed outside the study’s scope. Bearing these constraints in mind, researchers should be able to replicate the methods applied in this study across similar case studies and ensure accuracy within its limited scope. Potential research could be directed towards investigating the impact of either LAI, multiple reflections, or thermal conditions on the shade quality of children’s play areas, specifically in the context of UVR protection.

## 5. Conclusions

In an era of complex psychophysiological issues relating to children’s (a) increased digital indulgence vs. decreased physical activity, (b) greater involvement in non-competitive indoor sports vs. decreased outdoor excursions, and (c) less access to nature, the importance of outdoor recreation is being increasingly emphasized. Hence, the present research sheds light on an area foundational to both land use policy and the holistic health of future adults. While skin cancer prevention is considered external to urban open space design, shading has to date been treated as peripheral to children’s play space design in public park precincts. This research proposes a potential nexus between landscape design and a UVR protection framework for child-friendly Sun-safe Zones (SsZ). In order to reduce the risk of UVR exposure, this alternative methodology based on SsZ outlines several recommendations for an integrated shading strategy for children’s play areas in urban parks.

Among several recommendations propounded in the previous sections, this study concludes by recommending (a) a minimum canopy cover of 50% of the entire ground cover, (b) a minimum diameter for a shade (umbrella) of about 2.5 times the diameter of the table, and (c) an ideal umbrella height of 90 cm from the table surface. These recommendations on shading in urban parks and children’s play areas can be implemented beyond the context of the Perth region to other regions globally with similarly harsh environments and high UVI for the greater part of the year. The outcome of this research addressing the lack of shade amongst Perth’s play areas, which perpetuates unsustainable lifestyles by dissuading parents from bringing their children to unshaded outdoor venues, has the potential to influence SsZ designs for similar climatic geospatial contexts.

## Figures and Tables

**Figure 1 ijerph-20-00114-f001:**
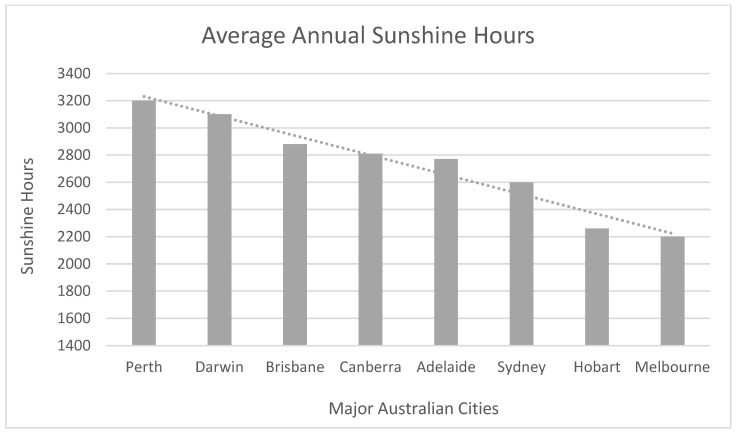
Average annual sunshine hours for major Australian cities (based on data from the Bureau of Meteorology, Australian Government).

**Figure 2 ijerph-20-00114-f002:**
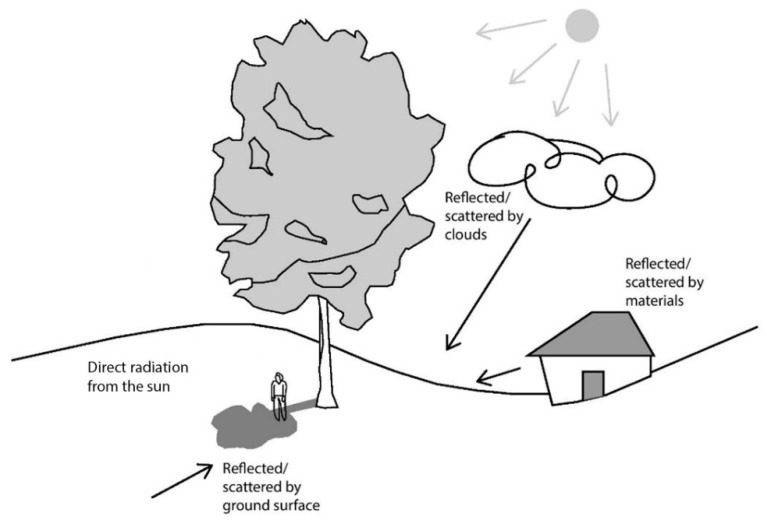
Incidence of indirect UVR.

**Figure 3 ijerph-20-00114-f003:**
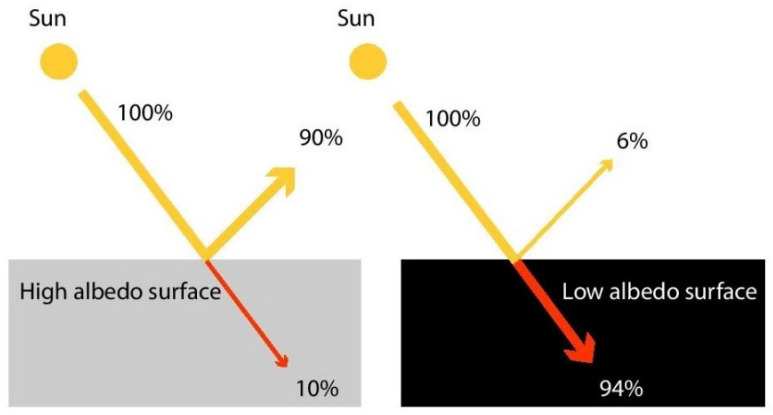
Albedo effect: the percentages of solar radiation indicated above as being reflected/absorbed are only indicative.

**Figure 4 ijerph-20-00114-f004:**
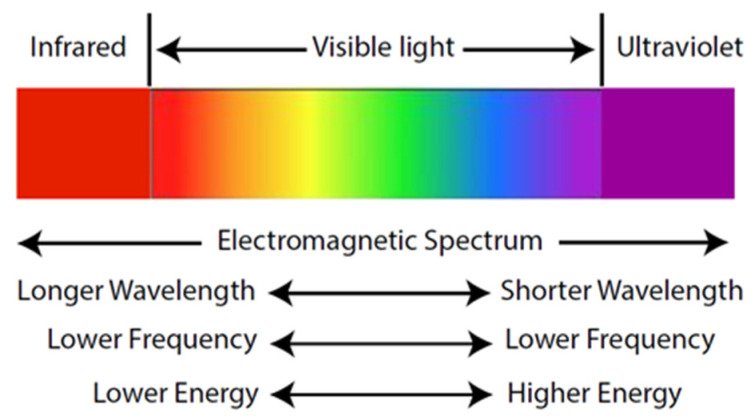
Electromagnetic spectrum of sunlight.

**Figure 5 ijerph-20-00114-f005:**
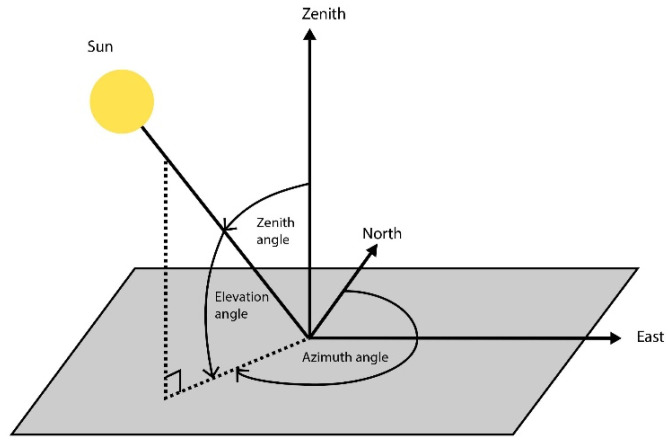
Representation of azimuth and zenith angles.

**Figure 6 ijerph-20-00114-f006:**
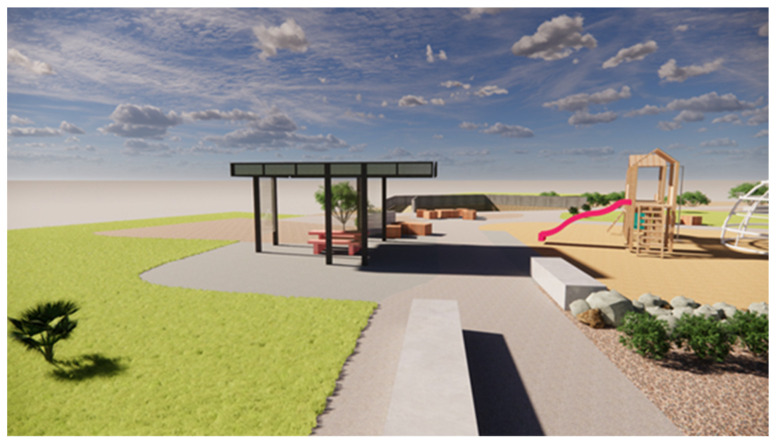
Shade structure at Baldivis One71 Pump Track.

**Figure 7 ijerph-20-00114-f007:**
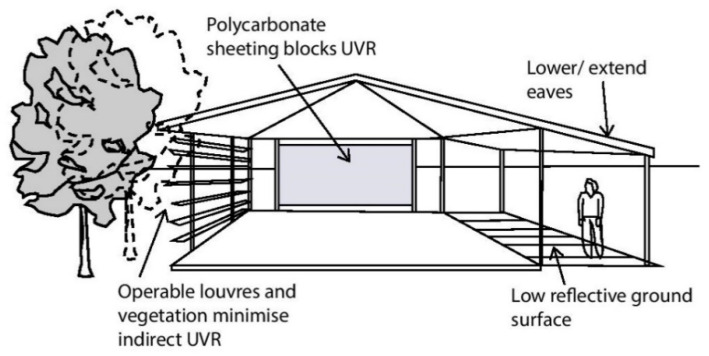
Strategies to minimize indirect UVR.

**Figure 8 ijerph-20-00114-f008:**
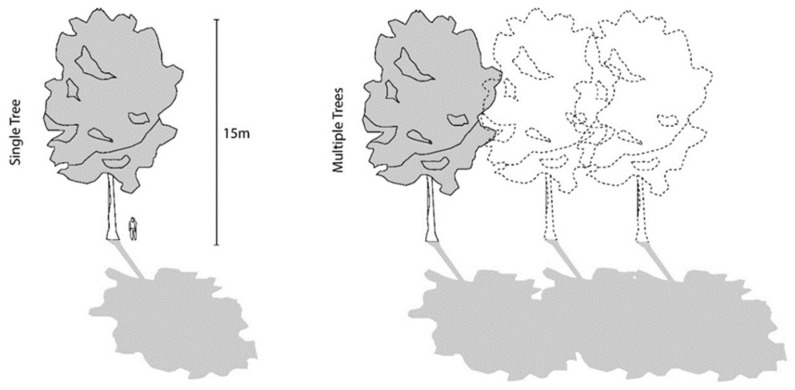
Trees planted with overlapping canopies.

**Figure 9 ijerph-20-00114-f009:**
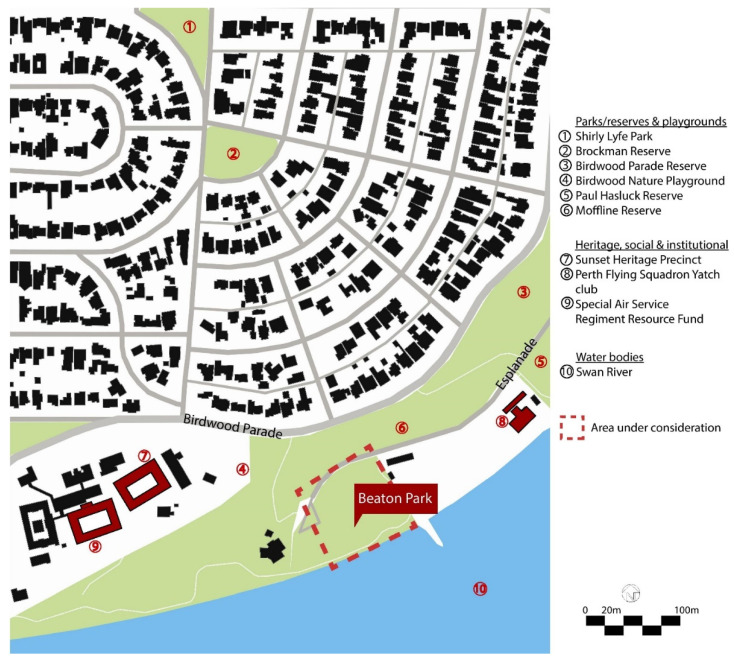
Contextual location.

**Figure 10 ijerph-20-00114-f010:**
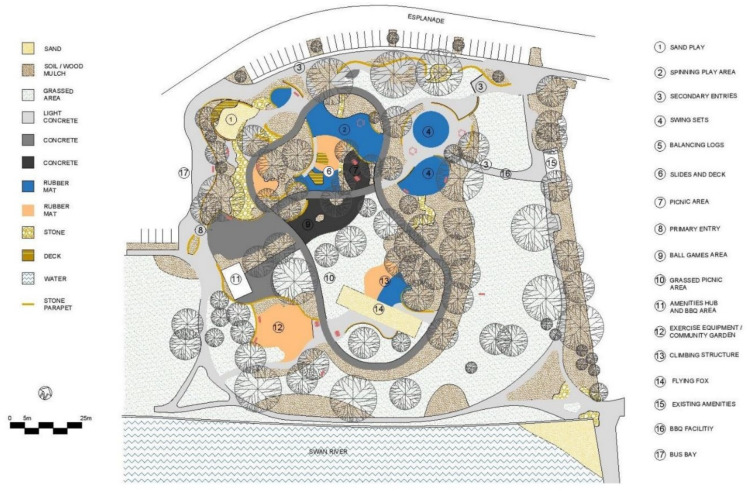
Site Plan.

**Figure 16 ijerph-20-00114-f016:**
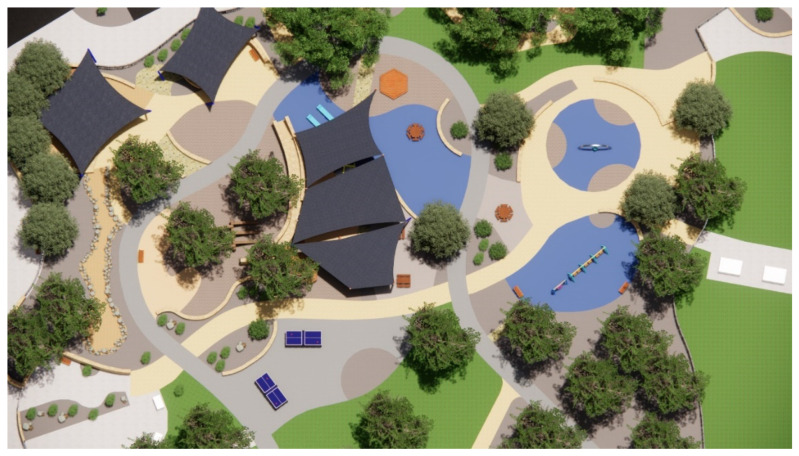
Shade sails, unshaded children’s play areas, and picnic tables during midday on the summer solstice.

**Figure 17 ijerph-20-00114-f017:**
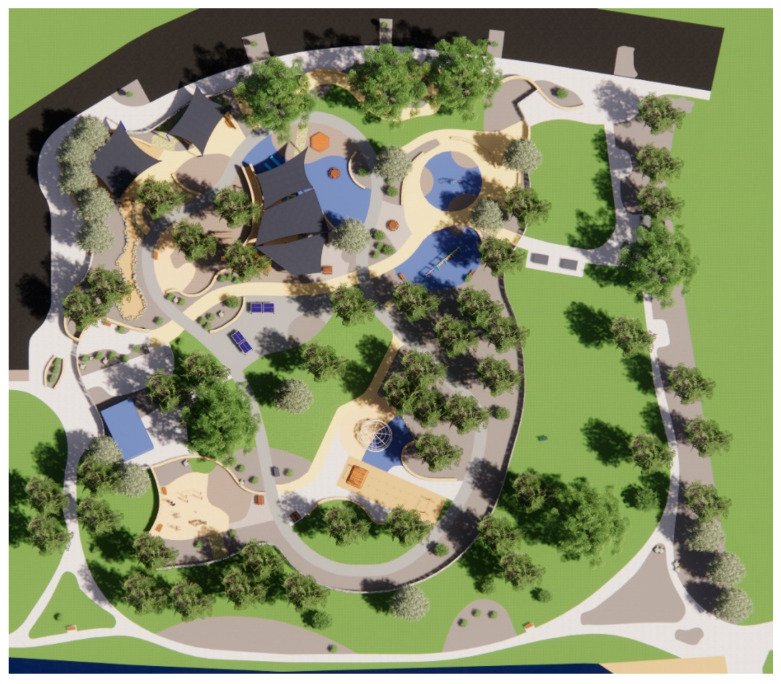
At 9 am on the summer solstice.

**Figure 18 ijerph-20-00114-f018:**
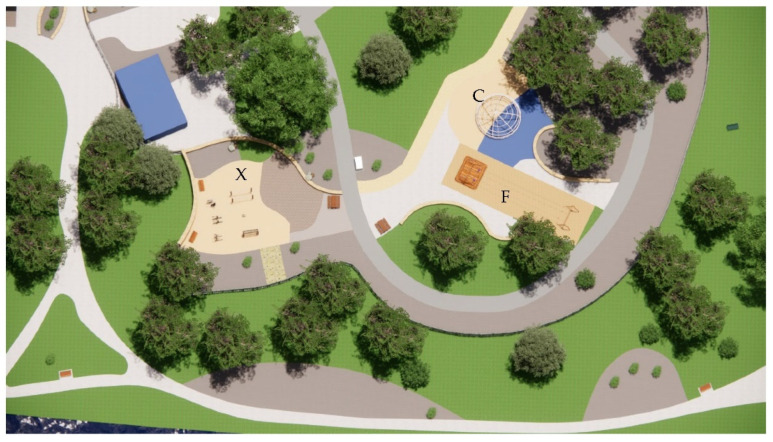
Flying fox, unshaded children’s play areas, and picnic tables during midday on the summer solstice.

**Figure 19 ijerph-20-00114-f019:**
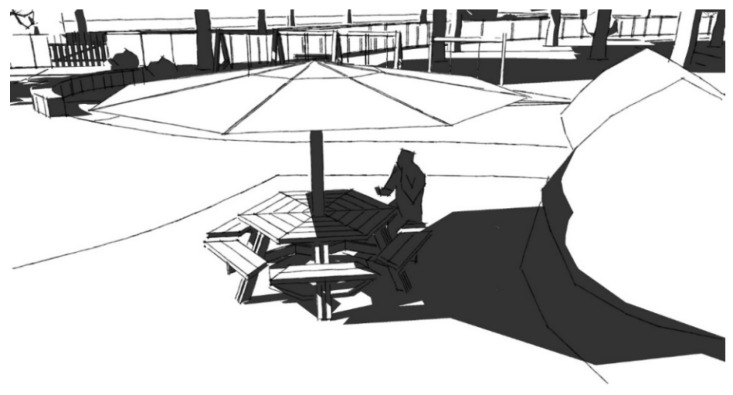
Picnic table at Beaton Park partially shaded by an umbrella at midday on the winter solstice.

**Figure 20 ijerph-20-00114-f020:**
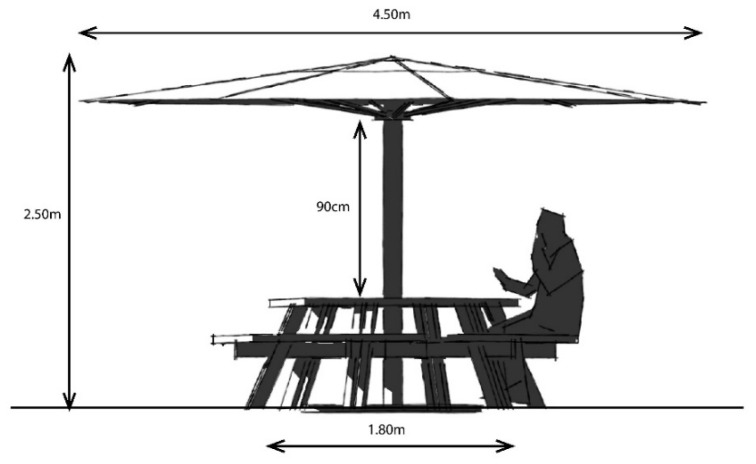
Dimensions of shade umbrella: umbrella diameter = 2.5 × diameter/diagonal length of table.

**Figure 21 ijerph-20-00114-f021:**
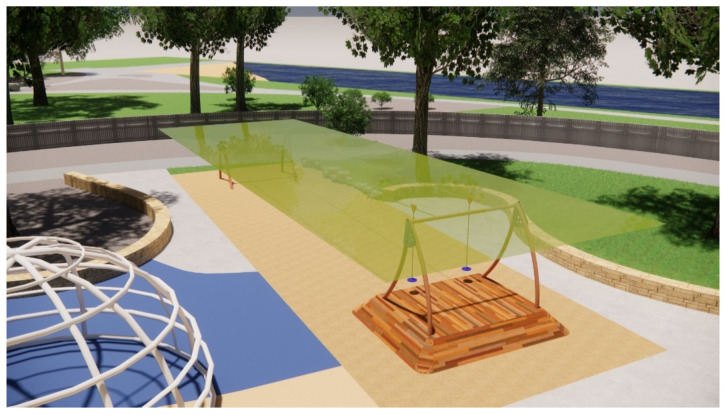
Flying Fox at Beaton Park fixed with tinted UVR-protected polycarbonate sheeting.

**Figure 22 ijerph-20-00114-f022:**
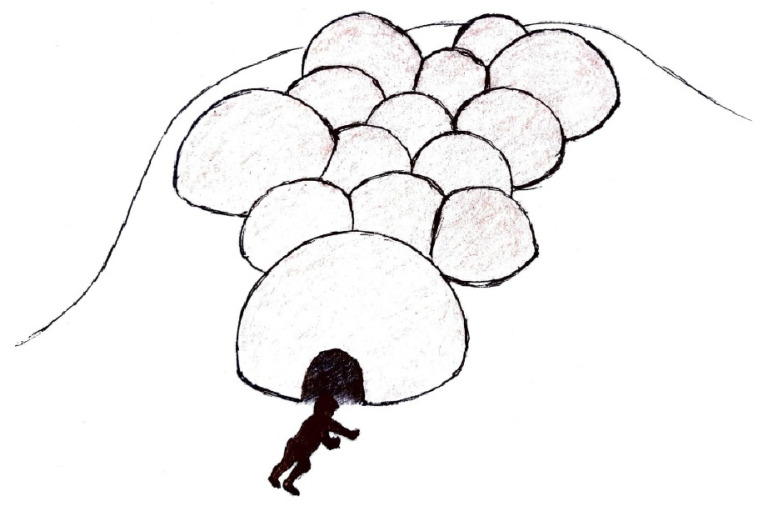
Playground mounds constructed from adobe—sketch.

**Figure 23 ijerph-20-00114-f023:**
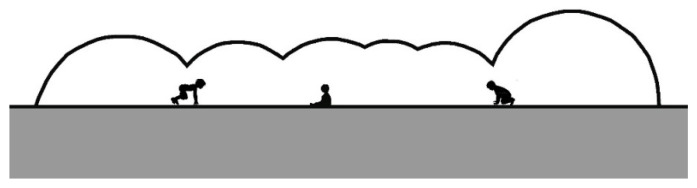
Playground mounds constructed from adobe—section.

**Figure 24 ijerph-20-00114-f024:**
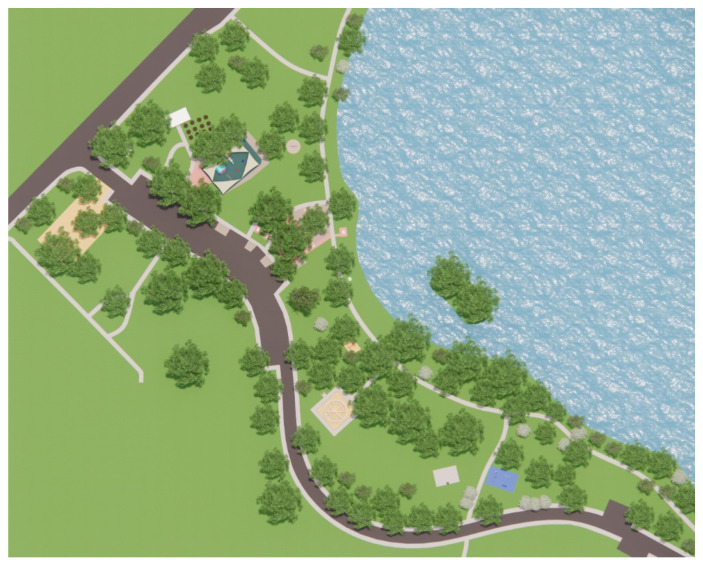
Tomato Lake Playground, Kewdale, at midday on the summer solstice.

**Table 1 ijerph-20-00114-t001:** Urban Metrics [34].

**Venue Name**	Jo Wheatley All Abilities Play Space (Beaton Park)
**Area**	2.4 ha
**Typology**	Large neighborhood park
**Local Government Authority (LGA)**	Dalkeith
**Population density (Belmont)**	14.5 persons/ha.
**Catchment distance**	800 m
**Catchment area**	100 ha
**Catchment population**	1450 persons

**Table 2 ijerph-20-00114-t002:** Temperature History (information supplied by Weatherzone based on data from the Bureau of Meteorology) [35].

**Mean Minimum Temperature**	8 °C (July/August), 18 °C (February)
**Mean Maximum Temperature**	17 °C (July), 30 °C (February)
**Hottest**	43.3 °C (4 February 2020)
**Coldest**	2.3 °C (19 May 2020)

**Table 3 ijerph-20-00114-t003:** Materials, their respective albedo values, and UPF offered [31].

Materials	Reflectivity/Albedo (%)	UVR Absorbed		Ultraviolet Protection Factor (UPF)
		%	Protection Score	
Dry sand	20%	80%	Low	5
Grassed lawn	2%	98%	Excellent	50
Shrubs	1%	99%	Excellent	50+
Open water	5%	95%	High	20
Dark concrete	10%	90%	Moderate	10
Light concrete	15%	85%	Low	7.5
Soil/wood mulch	5%	95%	High	20
Timber deck	5%	95%	High	20
Rubber mat	10%	90%	Moderate	10
Stone	20%	80%	Low	5
Asphalt road	5%	95%	High	20

**Table 4 ijerph-20-00114-t004:** Vegetation present [36,37].

*Botanical Name*	Common Name	Origin	Foliage	Height at Maturity (m)	Canopy Spread (m)	Time to Maturity (Years)
*Agonis flexuosa*	West Australian Weeping Peppermint	WA native	Evergreen	12–15	10	15–20
*Eucalyptus rudis*	Flooded Gum	WA native	Evergreen	10–20	8–12	10–20
*Eucalyptus cornuta*	Yate	WA native	Evergreen	10–20	8–12	10–20
*Eucalyptus leucoxylon*	White Ironbark	SA native	Evergreen	10–27	6–18	10–27
*Eucalyptus camaldulensis*	River Red Gum	AUS native	Evergreen	15–45	15–30	15–45
*Eucalyptus spathulata*	Swamp Mallee	WA native	Evergreen	6–12	6	10–12
*Tamarix aphylla*	Salt Cedar	Exotic	Evergreen	15–18	10–15	10–20
*Melaleuca armillaris*	Drooping Melaleuca	AUS native	Evergreen	5–10	5–10	5–10
*Melaleuca quinquenervia*	Paperbark	AUS native	Evergreen	8–15	5–10	15–25
*Melaleuca nesophila*	Pink Melaleuca	WA native	Evergreen	5–10	5–10	10–20
*Melaleuca squamea*	Swamp honey-myrtle	AUS native	Evergreen	2–6	1–3	5–10
*Melaleuca viminalis*	Weeping bottlebrush	AUS native	Evergreen	10	4	10–20
*Melaleuca linariifolia*	Flaxleaf Paperbark	East AUS native	Evergreen	6–9	4–7	10–20
*Callistemon salignus*	White Bottlebrush	AUS native	Evergreen	7	4	10–20
*Allocasuarina verticillata*	Drooping She-oak	AUS native	Evergreen	6–10	6–10	20–50
*Casuarina equisetifolia*	Ironwood	AUS native	Evergreen	12–20	6	12–20
*Casuarina cunninghamiana*	River Oak	AUS native	Evergreen	20	10	20–30

**Table 5 ijerph-20-00114-t005:** Site Visit data.

Visit	Day/Date	Time	Peak UVI	Temperature (Degrees Celsius)	No. of Users
1	Tuesday, 7 July 2020	4:00 pm	4 at 11:57 am	17–19	18–20
2	Tuesday, 29 September 2020	4:30 pm	5.4 at 12:08 pm	18–20	58–60
3	Wednesday, 30 September 2020	5:30 pm	4.1 at 1:55 pm	19–21	25–27

**Table 6 ijerph-20-00114-t006:** Analysis data.

**Area under Consideration**	2.4 ha
**Canopy cover**	0.65 ha
**Canopy percentage**	27% of the total ground cover
**Percentage of available shade (summer solstice at midday)**	29% of total ground cover

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
