# Peer review of "Sun-Safe Zones: Investigating Integrated Shading Strategies for Children’s Play Areas in Urban Parks"

_ijerph, 2022, doi:10.3390/ijerph20010114_

Round 1

Reviewer 1 Report (Previous Reviewer 1)

Thank you for taking my comments into consideration.

The paper is now well organized. 

Author Response

Thank you very much for your review and affirmation. 

Reviewer 2 Report (Previous Reviewer 3)

The authors have responded to review, and now do include some references to other park studies including more sophisticated efforts and including comfort factors. It is still questionable how this study adds significantly to the literature on this subject.  For example details of microclimate and comfort need to be explored in addition to the shade analysis. Shade analysis is important but the research edge is on additional parameters.

Author Response

Thank you very much for your comments.

The significance of the study in adding to existing literature has been further highlighted in the introduction as follows:

There have not been many published studies that explore UVR exposure and shade availability in children’s play areas. The two most comparable studies would be a) the “PlaSMa” study by Schneider et al. [18] that investigated 144 playgrounds in Mannheim, Germany, to establish via quantitative research methods a significant lack of shade in play areas, and b) a study by Gage et al. [19] across 50 New Zealand playgrounds to investigate the quantitative and qualitative shade availability in Wellington play areas to reveal insufficient shade. However, to our best knowledge, no prior studies have addressed both the quantitative and qualitative shade availability in relation to a children’s playground setting in Australia and proposed recommendations to enhance UVR protection.

Exploration of thermal comfort conditions has been deliberately left out of the study because of the narrow scope of the current research. The studies cited above for comparison have also discounted comfort conditions and other microclimate details. This was done so based on the suggestion of previous reviewers that advised:

"An article that describes one idea well is better than an article that tries to make too many conclusions from too many sources, surveys, tables and Figures. If the main idea of the article is intended to examine effective shade in children's play areas, then I suggest concentrating on just that aspect of the work".

Reviewer 3 Report (New Reviewer)

The paper under review examines an interesting topic: Integrated shading strategies for children’s play areas. While the paper provides a lot of useful information, it will benefit from some serious revisions. Below I discuss some areas for improvement.

1. While the paper is readable, it is ill-structured. For example, the Introduction reads like a literature review. You should introduce your research objectives and research questions as soon as possible rather than mentioning them after describing a long list of previous studies—in the introduction, you should only talk about some of the most relevant studies and, more importantly, the gaps in knowledge remaining in the literature. You can always add a stand-alone section for Literature Review.

2. The citation style is not consistent with the IJERPH style.

3. Terms such as UV, UVR, WA should be spelled out in their first appearance.

4. Figures:  what are sources of information?

5. Why do you think your stated contributions are really contributions? They are contributions in what way? In the current write-up, you only mentioned what you did, but you need to justify that they are indeed contributions to the literature--i.e., you have filled some gaps in the literature. Also, is it the first study of its kind? What kind? None of these is discussed.

6. Specific reasons for selecting your case studies should be provided.

7. More details on your method (e.g., "virtual park modeling") should be provided to better inform the reader. 

8. I am concerned that only a few parks were selected as your case studies, which prevents you from drawing broader generalizations of your results. For example, do your results only apply to some areas with very similar geographical conditions? Or do you have something more general to share with other areas? 

9. How are your findings compared to previous studies? 

10. Any implications for future studies?

Round 2

Reviewer 2 Report (Previous Reviewer 3)

authors have addressed concerns of this reviewer

Author Response

Thank you for all your feedback. 

Reviewer 3 Report (New Reviewer)

The paper has been much improved.

Author Response

Thank you for all your feedback. 

This manuscript is a resubmission of an earlier submission. The following is a list of the peer review reports and author responses from that submission.

Round 1

Reviewer 1 Report

In general, it's an excellent paper and deserves to be published, with minor revisions:

-       Figures 4, 5, 6, 8 need to be mentioned within the text in pages 5, 6, 7.

-        Line 307, “Case Study: Beaton Park, Dalkeith”, it’s better after the subtitle to start with text describing the case study prior to Figure10.

-        Many figures and tables starting from p. 11 need to be mentioned within the text.

Reviewer 2 Report

Overall this paper is more like a design-related master thesis/report rather than a journal article. It is not a research article and does not meet the quality of IJERPH. There’s no clear research questions or objectives.

The paper should be organized into: intro, method, results, conclusion and discussion. The introduction should talk about previous studies most related to your study (not common knowledge). The method should be specific. The results part is really vague. For this paper, it seems like the authors made some simulations. However, we didn’t see any results from the simulation. Discussion should talk about limitations of your study, future study, similarities or dissimilarities with previous studies, etc.

Overall, this paper is more like a report or site survey that landscape architects made before their design phase.

Line 21. Can you refer a newer/recent study instead of a 2008’s paper?

Figure 1. needs to adjust the size. Also ‘figure 1’ needs to be appear in your text.

Background – these are common knowledge to most academic researchers and readers of this journal. I don’t think this part is necessary. Review on recent studies closely related to your study can be much more helpful for you to define the gaps you are dealing with in this study.

There’s no clear statement of your methods.

Line 389. Need reference

Line 470. Why 50%?

Reviewer 3 Report

No doubt important topic. It tends to read like a planning report. How can other readers replicate the study and assure of its accuracy?

line 140 why specifically North Caroline Climate Office reference which is repeated throughout instead original of important science references?

line 144 statement should read with everything else being equal such as thermal admittances.

line 146 some researchers are designing darker surfaces which are highly reflective in the infrared offsetting heating.

line 150 use some important science references.

line 176 use original references

line 182 induce not precipitate

Fig. 4 source? scaling of temperature?

line 349 how does shade analysis work? what are methods? source? how does model account for multiple reflections?

There are many microclimate studies, what are some for comparison to your results?